# Investigation and Accounting Research of VOC in Daily and Specialty Ceramic Industry

**Yue Cheng** [1,*] **, Jiaxiang Jiang** [1] **, Lei Xia** [1] **, Hongxia Xu** [2] **, Changlin Ye** [3] **, Juan Sun** [3] **and Rui Gu** [4]

1. Department of Environmental Engineering, Jingdezhen Ceramic Institute, Jingdezhen 333403, China; jiangjiaxiang@foxmail.com (J.J.); rocky.xia@zeiss.com (L.X.)
2. Environmental Engineering Evaluation Center, Ministry of Environmental Protection, Beijing 100035, China; xuhx@acee.org.cn
3. Jiangxi Ecological Environment Monitoring Center, Nanchang 330077, China; xiaoyaofu2022@163.com (C.Y.); senjouye@163.com (J.S.)
4. Jiangxi Ronghui Environmental Protection Technology Co., Ltd., Nanchang 330006, China; estd_ray@sina.com
* Correspondence: 001025@jcu.edu.cn

**Abstract:** With the deepening of modernization in China, the situation of air environmental pollution in China is very serious, and the environmental capacity is becoming less and less. Haze has gradually become one of the main sources of pollution in most cities in China. Volatile organic compounds (VOC) have the activity of participating in photochemical reactions and can participate in a variety of complex reactions in the atmosphere to promote the formation of haze and ozone. Through the experiment of daily ceramic flower paper baking, it can be concluded that the burning loss of flower paper accounts for more than 60% of the total quality of flower paper, and most of them are between 60% and 80%. Among them, the burning loss of PVB flower paper is about 80%, the water transfer paper is about 60%, and the low temperature flower paper is about 86%. In the experiment, the VOC proportion of low temperature flower paper, the PVB paper and water transfer paper accounted for 3.17%, 0.92% and 0.45%, respectively. Through the wax removal experiment of special ceramics molded by hot pressing, it is found that paraffin wax, beeswax and oleic acid are used as binders, and the amount of VOC emitted is also different when the dosage range is different; in addition, the burning loss is generally between 10% and 20% in the process of wax removal. The content of VOC in flue gas is about 0.42%, accounting for 0.046% of special ceramics. It provides an important basis for the calculation of VOC generated during the process of ceramic flower baking.

**Keywords:** flower paper; bake; special ceramics; wax discharge; VOC; emission

## 1. Introduction

VOCs are important precursors for the formation of fine particulate matter ($PM_{2.5}$) and ozone ($O_3$) [1]. Compared with the control of particulate matter, sulfur dioxide and nitrogen oxides, China's VOCs have a weak management foundation and have become a short-board for atmospheric environmental management. At present, petrochemical, chemical, industrial coating, packaging and printing, oil storage, transportation and sales industries have become the key sources of VOCs in China. In order to further improve the environmental air quality, it is urgent to comprehensively strengthen the management of VOCs in key industries [2,3].

China is a big country in terms of ceramic production, and its production of ceramics for construction, sanitation and daily use has ranked the country first in the world for many years. The 2017 statistics show that the total output of China's ceramic products is about 276.53 million tons. Among them, the annual output of building ceramics is 10.15 billion square meters, accounting for nearly 70% of the world's total output; the annual output of sanitary ceramics is 218 million pieces, accounting for 50% of the world's total output; the annual output of daily-use ceramics is more than 40 billion pieces (the pieces refer to



complete functional ceramics), accounting for 60% of the world's total output; furnishings and art porcelain account for 65% of the world, and specialty ceramics account for 45% of the world's production [4–6].

There are more than 3600 enterprises above designated size in my country's ceramic product manufacturing industry, and tens of thousands of enterprises below designated size. Among the enterprises above designated size, architectural ceramics accounted for a large proportion, followed by sanitary ceramics and daily-use ceramics; below-scale display art ceramics accounted for a large proportion. Consistent with the raw material base and market, there have formed the main ceramic producing areas such as Foshan and Chaozhou in Guangdong, Jinjiang and Dehua in Fujian, Jingdezhen and Gao'an in Jiangxi, Faku in Liaoning, Jiajiang in Sichuan, Beiliu in Guangxi, Yixing in Jiangsu, Zibo in Shandong, Tangshan in Hebei, etc. Among them, the provinces south of the Yangtze River account for about 70% of the total output. Among them, Guangdong Province ranks first in architectural ceramics, sanitary ceramics, and daily-use ceramics, and Jiangxi Province ranks first in special ceramics. However, the scale of a single enterprise is not large, and the industrial concentration is low. The output of the top 10 enterprises in construction and sanitary ceramics accounts for 13% and 17% of the entire industry, respectively, while other ceramics are even lower. There are about 3350 building ceramics production lines, with a total production capacity of 14 billion square meters; there are also more than 200 sanitary ceramics tunnel kiln production lines and more than 1000 shuttle kilns.

Among them, special ceramics (also known as "industrial ceramics", "engineering ceramics", "modern ceramics", "fine ceramics" and "high-tech ceramics") are various new types of ceramics that have emerged with the development of modern industrial technology. The general term ceramics is widely used in chemical, metallurgy, machinery, electronics, energy and other high-tech fields. Among them, the number of enterprises in Pingxiang City, Jiangxi Province is the largest in the country.

According to the survey, the industries that emit VOCs in the product process in industrial sources mainly include the following: refining, chemical industry (such as medicine, etc.), coating, rubber and plastic products, printing and packaging, textile printing and dyeing and furniture (such as wood-based panel production, etc.), shoemaking, etc.; these belong to the key industries of VOCs emission.

There are multiple VOCs emission links in the ceramic production process. Most architectural ceramics companies have built inkjet printing lines, and the ceramic inks used contain surfactants and organic solvents, which emit VOCs during use. In the roasting process of daily-use ceramics, the burning of the PVC components in the decals will emit VOCs. The use of organic solvents (paraffin wax, tung oil, etc.) in the wax removal and plastic removal processes of special ceramic production also has the problem of VOCs emission. The proportion of VOCs emitted by the ceramic industry is relatively small compared to other industries.

From the perspective of production scale, in addition to architectural ceramics, the production scale of sanitary ceramics, daily-use ceramics, furnishings art ceramics, special ceramics and other ceramic varieties are relatively small. Ceramic products can be used as decorative materials, functional materials and structural materials. The variety of colors and colors is diverse, and the performance is diverse and widely used. Different ceramic materials and product types have different maximum firing temperatures and firing times, which affect energy consumption and pollution emission levels [7].

In the production of daily-use ceramics, it involves the process of decaling and baking. The decals are mainly composed of screen printing inks (varnish and cover oil) or PVB (polyvinyl butyral) resins and inorganic pigments. In the decal process, the paper is generally wetted with water and attached to the surface of the ceramic glaze. Volatile organic matter is not produced during the decaling process; during the baking process, the varnish, cover oil or PVB resin decomposes to produce VOC [8,9].

When the production of special ceramics adopts the hot die casting process, in order to facilitate the shaping, it is necessary to use paraffin wax (without chlorine) to configure

the wax cake [10,11]. The hot-pressed green body needs to be waxed before calcination, and the paraffin wax in the blank is completely volatilized [12,13]. In order to prevent the deformation and perforation of the green body during the high-temperature melting and volatilization of the paraffin wax, the blank body needs to be loaded into a special one. In the middle of the raft, the alumina powder is covered to cover the blank to achieve the filling and supporting action during the wax discharge, and the blank is not deformed. The wax discharge means that the paraffin in the green body is melted and volatilized by high temperature (that is, a low-temperature sintering process is carried out in a firing kiln). Paraffin is a mixture of solid higher alkanes [14–16]. The main component has the formula $C_nH_{2n+2}$, where $n$ = 17–35. The paraffin wax melts completely at >100 °C, and the amount of volatilization increases. The paraffin wax is basically complete during the preheating in the low temperature zone. Volatilized into paraffin vapor, paraffin vapor is mainly a higher alkane, which is VOC [17].

Research should be carried out on the characteristics of VOCs emissions in the ceramic industry, to find out the production and discharge links of VOCs, and to clarify the types and concentrations of VOCs emissions. Focus should be placed on the inkjet printing process of architectural ceramics, the baking process of daily ceramics, the wax removal and plastic removal processes of special ceramics. Combined with the current situation of VOCs emission control in the ceramic industry and the existing feasible VOCs pollution control technologies, the feasible technical route for pollution control should be clarified, the VOCs pollution control policy in the ceramic industry should be formulated, and the VOCs treatment should be standardized in terms of the use of raw and auxiliary materials, the collection of VOCs, and the selection of treatment technologies.

The roasting process produced by daily-use ceramics and the wax-extracting process of special ceramics production will produce VOC, which will pollute the environment. The research on VOC volatilization in the production of daily-use ceramics and special ceramics has not been reported or reported so far. It is vital and meaningful to investigate and account for VOC in the ceramic industry. It will help to improve the scientific and accuracy of environmental impact assessment and prediction, standardize the environmental impact assessment of pollutants in construction projects, and further improve the technical guidelines system for environmental impact assessment. The purpose and outcome of the study will be used for emission inventory development or to update the emission inventory.

## 2. Experimental Section

### 2.1. Experimental Process

In the flower paper baking experiment, a number of different types of flower paper were weighted (FA1004B Electronic balance, Shanghai Precision Scientific Instrument Co., Ltd., Shanghai, China) and placed in a muffle furnace (WS-10-13 Jiangsu Dingshan Electric Protection Factory, Nanjing, China) and baked at 200–800 °C for half an hour. The relationship between the loss on ignition of the flower paper at different temperatures was calculated. Secondly, the VOC in the flue gas during the roasting process was pre-estimated by studying the TG/DTA (STA449C integrated thermal analyzer, Netzsch, Germany) and FT-IR (Nicolet 5700, American Thermoelectric, Waltham, MA, USA) map of the flower paper and each component. Finally, some Venezuela flue gas was sampled (KC-6120, atmospheric sampler, Qingdao Laoshan Electronic Instrument Factory Co., Ltd., Qingdao, China) and enriched (VOC solid absorption column) and sent to Jiangxi Mengbaomei (MBM) Environmental Testing Technology Co., Ltd. (Nanchang, China) for further analysis of VOC (GC (Clarus 680) and MS (SQ8 T)) in roasted flue gas. Finally, the VOC content in the flue gas during the baking process is determined. Because papers used can be thin or thick, the paper material is also various, and the ink on the flower paper contains organic and inorganic chemicals, so this study only studies some typical flower papers.

The special ceramic waxing experiment was sampled and placed in a muffle furnace and slowly heated to 800 °C. The flue gas was sampled to Jiangxi Province MBM Environmental Testing Technology Co., Ltd. to determine the VOC and determine the VOC

content in the flue gas. In addition to the special ceramic barbecue sampling, infrared spectroscopy and differential thermal gravimetric analysis of the components of special ceramics, paraffin, beeswax and oleic acid were also carried out. The types of experimental samples and sampling conditions are shown in Table 1.

**Table 1.** Test sample types and sampling conditions of roasted flower and waxing.

| No. | Name | Types | Physical Photo | Main Ingredient | Baking Temperature (°C) | Sampling Conditions | | |
|-----|------|-------|----------------|-----------------|------------------------|---------|---------|---------|
| | | | | | | T (°C) | P (hpa) | V (m³) |
| 1 | 95 ceramics | Special ceramics | | 95%$Al_2O_3$ + Paraffin wax + peak wax + oleic acid | 800 | | | |
| 2 | Side flower | water transfer paper | | Cover oil + varnish + ceramic pigment | 800 | | | |
| 3 | Fen wine | Low temperature flower paper | | (Cover oil + varnish + ceramic pigment) | 200 | | | |
| 4 | Tartar oil | / | | Vegetable oil or synthetic resin | 500 | 22–24 | 1000 | 95 |
| 5 | Cover oil | / | | Acrylic resin | 500 | | | |
| 6 | Social pig | PVB Paper | | ((polyvinyl butyral)) + pigment + varnish | 800 | | | |

Notes: (1) Water transfer paper, also known as small film paper, consists of the bottom paper (also known as paper base), the middle layer of sol (toner oil and pigment), and the surface cover oil. Paper ingredients, pigment (metal oxide), acrylic resin: 35–40%, aromatic solvent: 55–60%, additive: 3–8%. (2) Varnish, polymerize vegetable oil to A synthetic resin is added to a certain viscosity or added thereto to prepare it to have an appropriate viscosity. The main purpose is to adjust the viscosity or thickness of the ink. The usage of Jingdezhen is estimated to be 600–1000 T/a. (3) Cover oil, The ceramic flower paper cover oil is mainly made of thermoplastic acrylic resin, and the estimated dosage of Jingdezhen is 3000–4000 T/a. (4) PVB paper, mainly made of polyvinyl butyral as a raw material, is made into a film, and a pattern is printed on the surface of the backing paper, and can be used as a ceramic flower paper after printing. After baking the flowers, the PVB film is decomposed, leaving only the pigment on the ceramic surface.

### 2.2. Data Processing

(1) Based on the calculation of the loss on the baking (roasting) process of the daily-use ceramic flower paper, several kinds of flower papers were selected for baking, and the balance calculation method was also used to determine the loss on the paper during the baking process. The baking flower experiment first studied the heat loss and infrared spectrum of the flower paper and each component, and then studied the loss of the paper at different temperatures during the baking process. Finally, the VOC of smoke sample in roasted flue gas was analyzed by Jiangxi Mengbaomei (MBM) Environmental Testing Technology Co., Ltd. Finally, the VOC content in the flue gas of the roasted flower is determined. Because the paper is thin and thick, the paper material is also various. The ink on the flower paper contains organic and inorganic chemicals. Therefore, this study only studies some typical water transfer paper and PVB paper. The analysis measures the amount of VOC that is volatilized into the atmosphere.

$$\text{Calculation number}: m_3 = m_1 - m_2 \tag{1}$$

$m_1$: the weight of the flower paper before baking, g
$m_2$: the weight after baking, g
$m_3$: the weight of the loss, g

(2) The body of the special ceramics produced by hot die casting contains a certain proportion of paraffin, beeswax and oleic acid. If the amount of VOC volatilized into the atmosphere during heating is known, the paraffin in the special ceramic ingot can also be used first. Beeswax and oleic acid were subjected to Fourier infrared analysis and differential thermal gravimetric analysis. In the experimental process, we first studied the loss of ignition during the entire process of wax removal and then sent samples to determine VOC. The experimental protocol is the same as the baking experiment.

$$\text{Calculation number}: \ m_3 = m_1 - m_2 \tag{2}$$

$m_1$: the weight of the special ceramic wax before the wax, g
$m_2$: the weight after waxing, g
$m_3$: loss of special ceramics, g

(3) VOC detection method: VOC in the flue gas is collected into six VOC absorption tubes by atmospheric sampler through the baking and waxing experiments, and these samples are analyzed by Jiangxi MBM Environmental Testing Technology Co., Ltd., through GC-MS (gas chromatography-mass spectrometry) qualitative and quantitative analysis of VOCs in the absorption tube. The test is based on the determination of VOC in fixed sources of waste gas by solid phase adsorption—thermal desorption/gas chromatography—mass spectrometry (HJ734-2014, National Environmental Protection Standard of the People's Republic of China, Beijing, China). In the practical application, the detection technology can realize qualitative and quantitative analysis for unknown gases. The detection technology is relatively mature and is widely used to detect VOC components in the air. In general, the detection of volatile organic compounds by mass spectrometry can limit their detection specifications to 1~10 μg/kg, which can accurately detect the actual content and type of organic matter, and can provide researchers with accurate data information [12].

## 3. Results and Discussion

### 3.1. FT-IR Analysis

3.1.1. Low Temperature Paper

Figure 1 is an infrared spectrum of a low temperature paper. The infrared spectrum of the water transfer paper can have a distinct –COOH absorption peak around 2096 cm$^{-1}$, and also exhibits an absorption peak of C=C at 1718 cm$^{-1}$, at 2955 cm$^{-1}$, 2934 cm$^{-1}$ and 2867 cm$^{-1}$ is the three adsorption peaks of –CHO vibration peak. The two absorption peaks around 1465 cm$^{-1}$ are the benzene ring vibration peaks, and the absorption peak at 743 cm$^{-1}$ is the mono substituted benzene ring vibration peak. The vibration peak near 1060 cm$^{-1}$ is a –CO-absorption peak. It is seen from the analysis that the low temperature paper film contains a plurality of benzene ring structures, and at the same time, multiple –CHO, C=C and C=O, will become VOC during the low temperature baking process at 180–200 °C.

3.1.2. Water Transfer Paper and PVB Paper

Figure 2 shows the infrared spectrum of water-shifted paper (a) and PVB paper (b). Figure 2a shows water-shifted paper, and Figure 2b shows PVB paper. By the infrared spectrum of the water transfer paper, there is a distinct –COOH absorption peak around 2096 cm$^{-1}$, and at the same time, the absorption peak of C=C is also exhibited at 1727 cm$^{-1}$. In the PVB spectrum on the right, the two absorption peaks at 3064 and 3030 cm$^{-1}$ are benzene ring C–H vibration peaks. The two absorption peaks around 1490 cm$^{-1}$ are benzene ring vibration peaks, and the absorption peaks at 755 and 700 cm$^{-1}$ are single-substituted benzene ring vibration peaks. The vibration peak near 1045 cm$^{-1}$ is a C=O

absorption peak. It is seen from the analysis that the PVB film contains a plurality of benzene ring structures while a plurality of –OH are substituted.

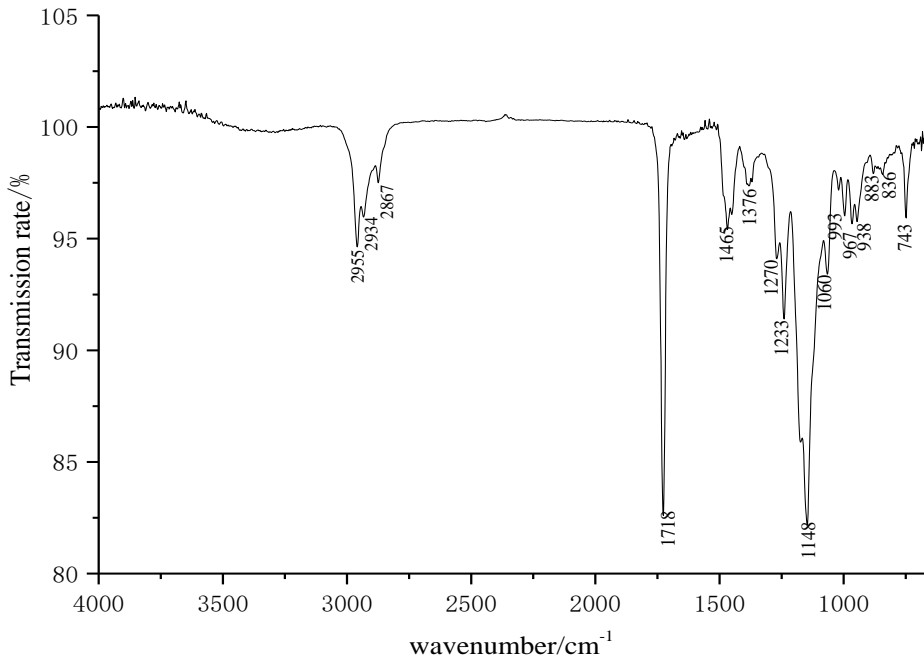

**Figure 1.** FT-IR analysis of low temperature flower paper.

### 3.1.3. Cover Oil and Varnish in Water Transfer Paper Components

Figure 3 shows the spectrum of the cover oil and the varnish. It can be concluded from the spectrum that each of the two substances has a strong functional group peak, and the number of absorption peaks is large, so that the molecular structure of the two substances is complicated. Both spectra at 2958 and 2730 cm$^{-1}$ have strong –CHO absorption peaks, indicating that there are many –CHO in the two species. At the same time, both substances showed a strong –OH absorption peak near 3540 cm$^{-1}$, indicating that a large amount of –OH was contained in the molecule. Both of these functional groups may be thermally decomposed into $CO_2$ and $H_2O$ during heating. The cover oil also shows the absorption peak of C=C near 1386 cm$^{-1}$, and it can be inferred that the cover oil is completely heated and requires a higher temperature.

### 3.1.4. Beeswax, Oleic Acid and Paraffin in Special Ceramics Group

Figure 4 is an infrared spectrum of beeswax, paraffin and oleic acid. From the infrared spectrum, it can be seen that the organic matter of beeswax and paraffin is more complex than the oleic acid organic substance, and the beeswax exhibits a strong –CHO at 2848 and 2915 cm$^{-1}$. The absorption peak, while observing that the paraffin component also exhibited a –CHO absorption peak at the corresponding 2848 and 2915 cm$^{-1}$. In addition, beeswax has a distinct absorption peak of C=O at 1735 cm$^{-1}$. The analogy is similar to that of beeswax and paraffin. Oleic acid is more complicated than beeswax and paraffin. It can be seen from the infrared spectrum that oleic acid shows a carboxyl-COOH peak at 1708, 2674 and 2935 cm$^{-1}$, and it is known that oleic acid is acidic. At the same time, the absorption peak of –CHO was found at 937 and 2854 cm$^{-1}$. C=C absorption peak is exhibited at 3008 cm$^{-1}$, which indicates that oleic acid requires a higher temperature to decompose completely.

### 3.2. DTA/TG Analysis

#### 3.2.1. Low Temperature Flower Paper

Figure 5 DTA/TG analysis of low temperature flower paper, two absorption curves and an exothermic curve of the low temperature paper DTA curve can be observed from

Figure 5. It exhibits an exothermic reaction between 360 and 388 °C, which is related to the exothermic decomposition of organic acids in the low temperature flower paper with –COOH and C=O functional groups. The organic substance decomposed by the organic substance having a benzene ring or decomposed by the resin during the heating process is decomposed again at about 502 °C. The TG curve began to decrease at 250 °C, indicating that the volatiles and decomposed substances in the flower paper began to decrease slowly. When the temperature was 388 °C, the heat loss reached a maximum, and then the weight was constant. In the actual baking process, the baking temperature of the low temperature flower paper is controlled at 180–200 °C, and a lot of organic matter is decomposed into VOC. Therefore, it can be inferred that the VOC concentration in the flue gas of the low temperature flower paper baking process will be higher.

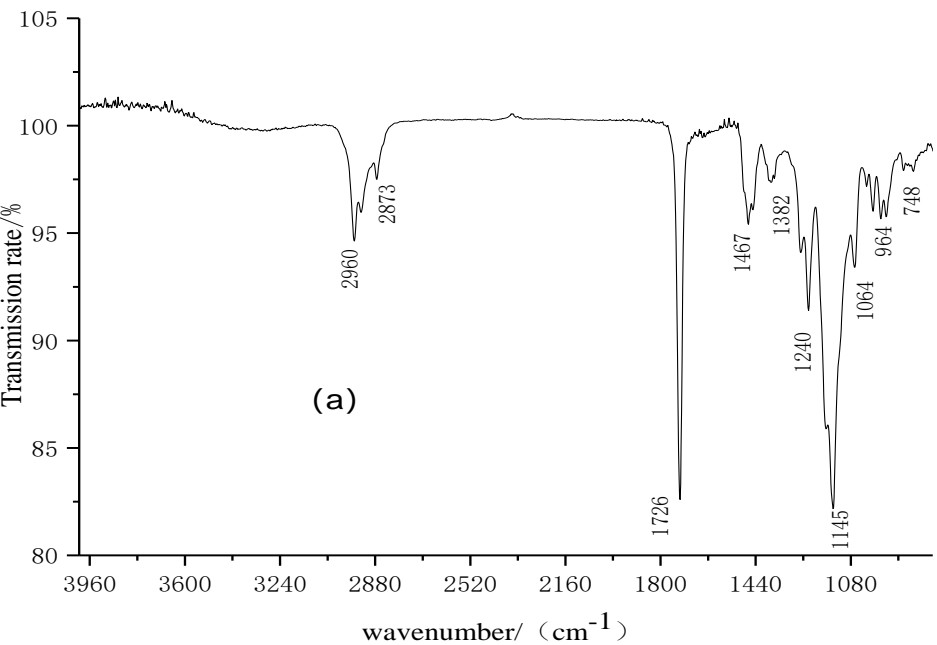

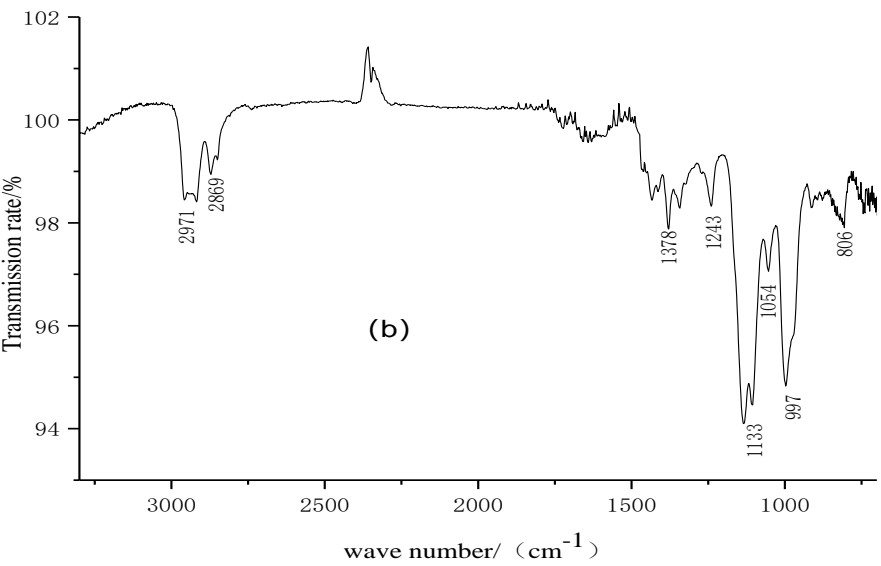

**Figure 2.** FT-IR analysis of water transfer paper (**a**) and PVB flower paper (**b**).

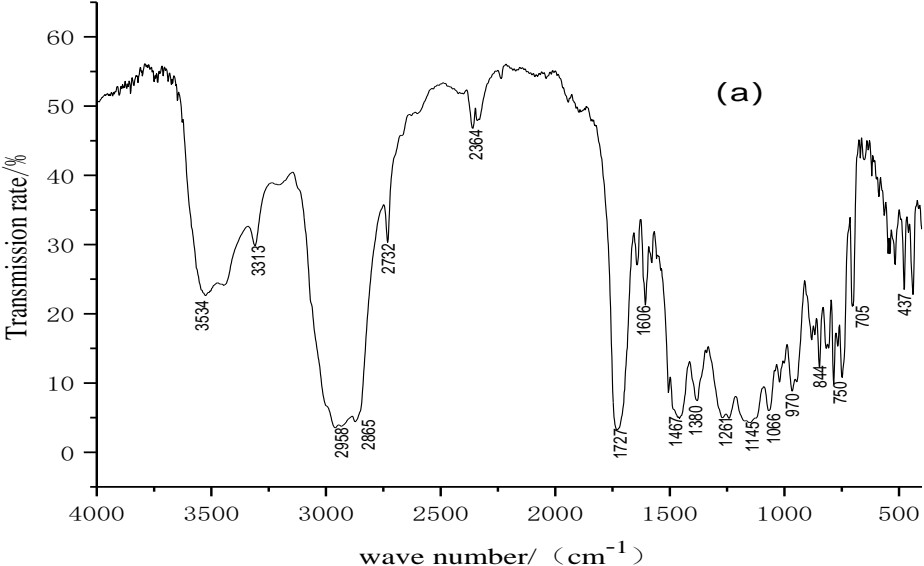

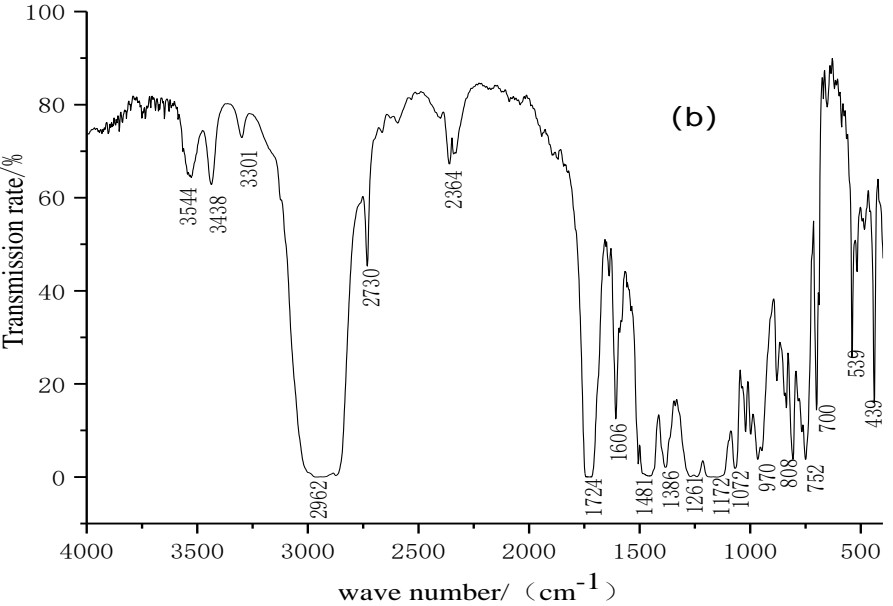

**Figure 3.** FT-IR analysis of cover oil (**a**) and varnish (**b**).

### 3.2.2. Water Transfer Paper and PVB Paper

Figure 6 is the DTA/TG of water transfer paper (a) and PVB paper (b). Figure 6a shows two absorption curves and an exothermic curve of the water transfer paper DTA curve. It exhibits an exothermic reaction between 304 and 388 °C, which is related to the exothermic decomposition of organic compounds with –COOH and C=O functional groups in water transfer. The TG curve began to decrease at 150 °C, indicating that the volatiles and decomposed substances in the flower paper began to decrease slowly, and the heat loss reached a maximum at 388 °C. The weight is then constant, indicating that a high temperature resistant inorganic pigment remains. Figure 6b shows that there are multiple absorption peaks and exothermic peaks, which are related to the functional group organics with large differences in PVB films. The exothermic reaction before 387 °C may be the decomposition of organic matter with –OH, and the decomposition of the organic matter with the benzene ring at about 507 °C or the decomposition of the organic resin

during the previous heating process. It can be speculated that the VOC concentration in the low-temperature roasted flue gas will be higher during the actual roasting process.

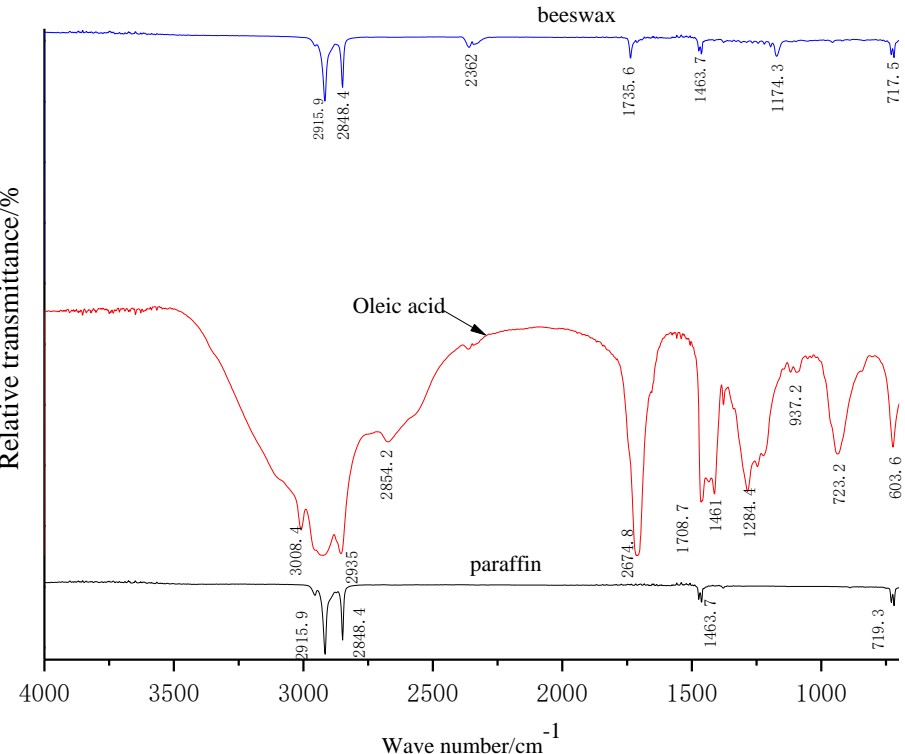

**Figure 4.** FT-IR analysis of beeswax, oleic acid and paraffin.

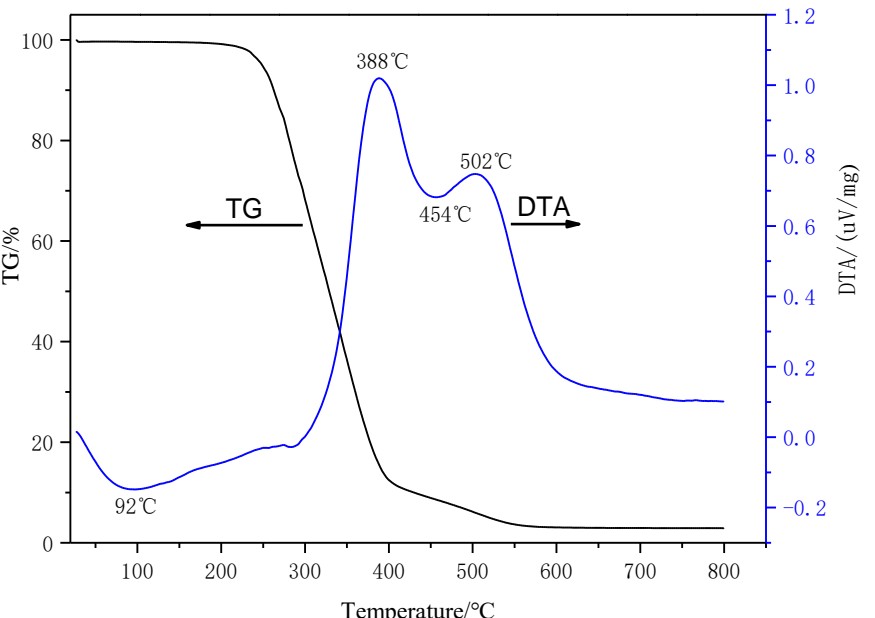

**Figure 5.** DTA/TG analysis of low temperature flower paper.

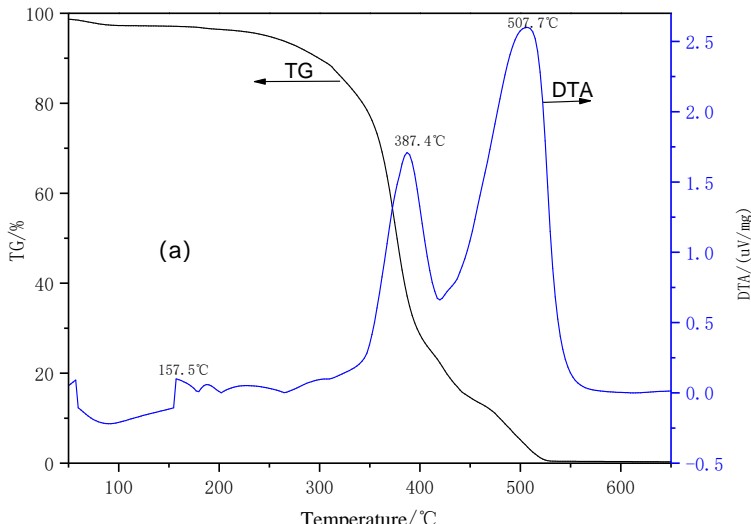

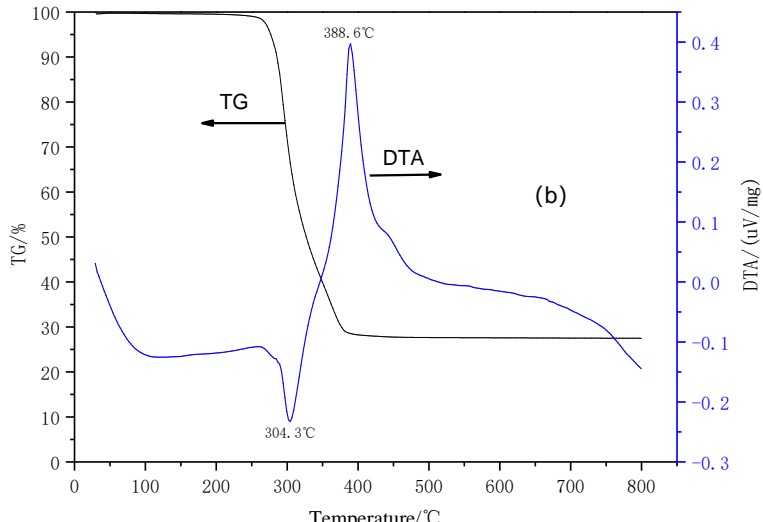

**Figure 6.** DTA/TG analysis of water transfer paper (**a**) and PVB paper (**b**).

### 3.2.3. Water Transfer Paper Component Cover Oil and Varnish

Figure 7 is the DTA/TG of cover oil (a) and varnish (b). DTA curve in Figure 7 shows that multiple reactions occur in both materials during heating. In the DTA curve of Figure 7a, at −188 °C, the –OH organic matter begins to decompose. At 291 °C, C=O and –CHO begin to decompose and some of the resin begins to decompose to produce small molecular organic matter, reaching 428 °C at temperature. When the heat-decomposable substance in the cover oil is substantially decomposed, the TG curve of Figure 7a tends to be horizontal at this temperature. In Figure 7b, the whole process of thermal decomposition in the DTA curve is roughly similar to that of the cover oil. The absorption curve at around 213 °C is that the organic matter with –COOH and –OH begins to be thermally decomposed. At about 293 °C, a small amount of macromolecular resin begins break down. After a plurality of endothermic exothermic processes, the TG curve tends to be horizontal after 550 °C, and the substances susceptible to thermal decomposition in the varnish are also substantially decomposed.

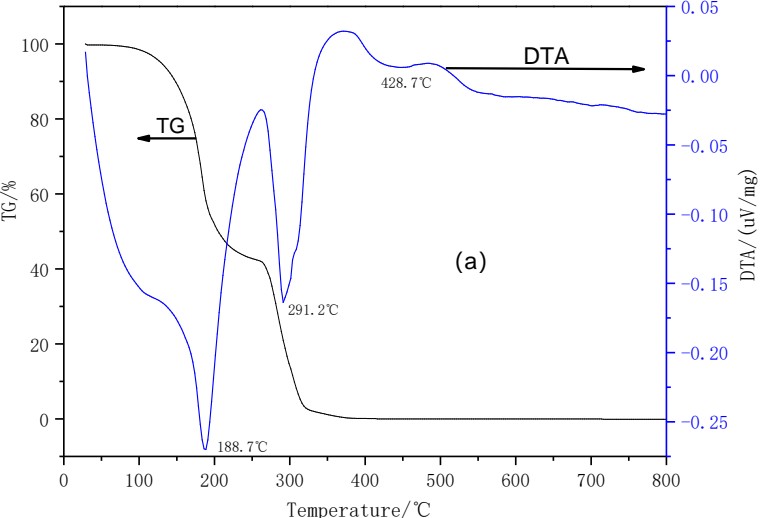

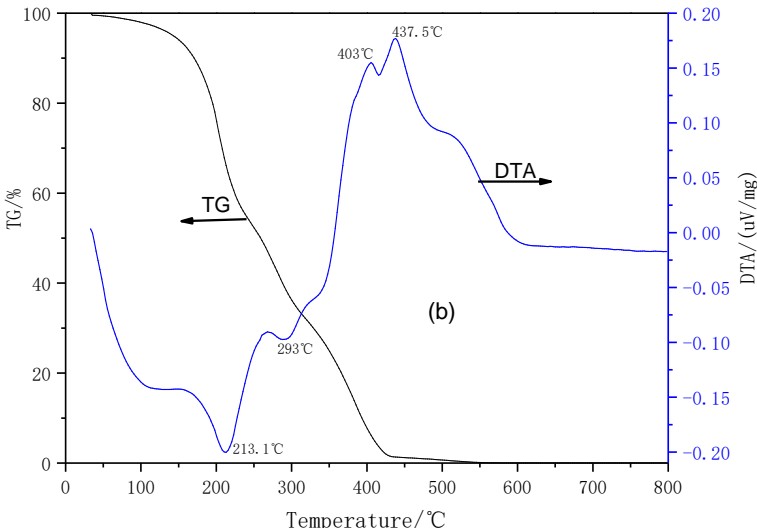

**Figure 7.** DTA/TG analysis of cover oil (**a**) and varnish (**b**).

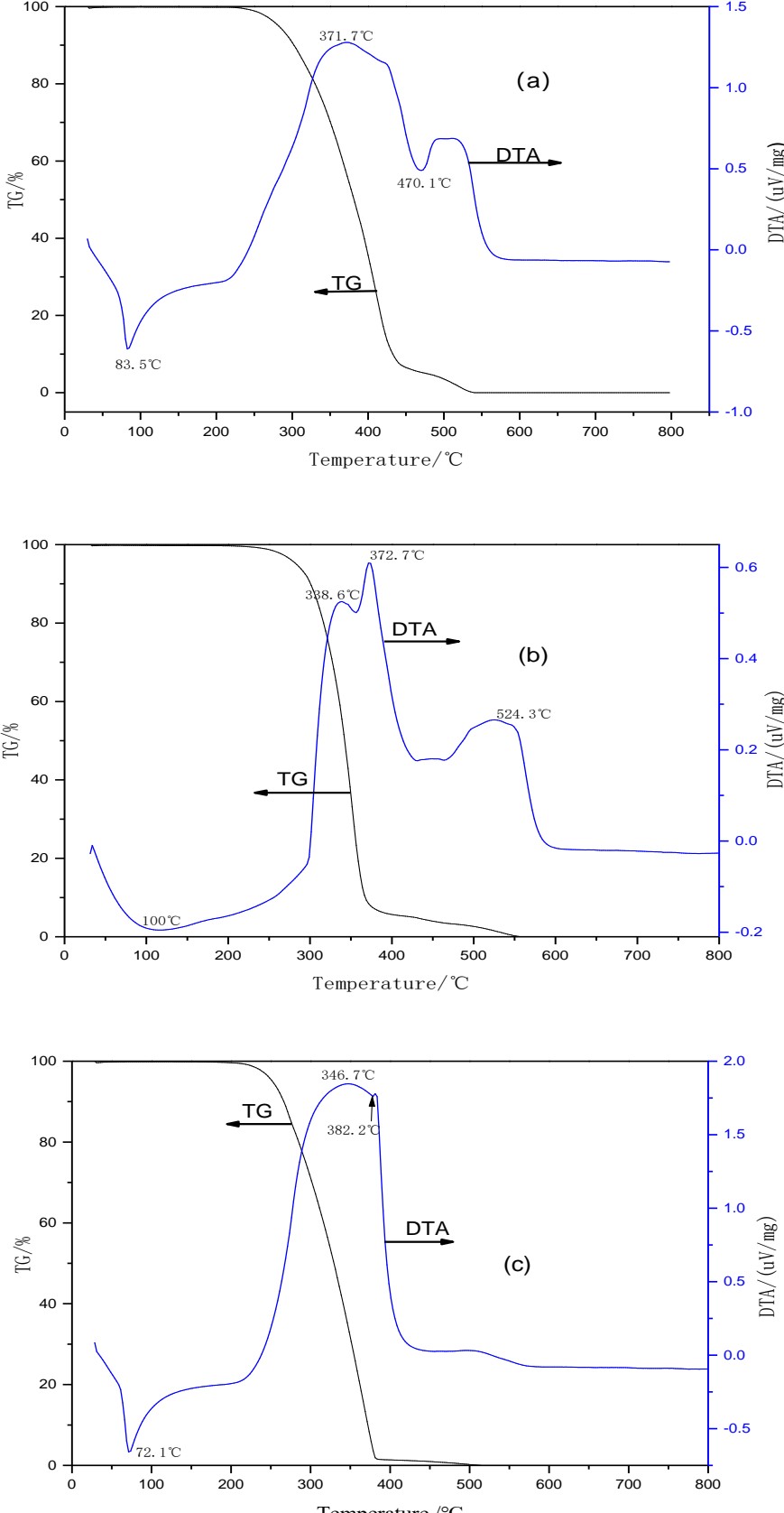

**Figure 8.** DTA/TG analysis of beeswax (**a**), oleic acid (**b**) and paraffin (**c**).

*3.3. Effect of Temperature on the Loss of Paper Waste*

3.3.1. Loss on Ignition of the Same Flower Paper at Different Temperatures

The side flower water transfer paper sample was placed on the porcelain plate after high-temperature baking to determine the initial weight of the flower paper, and then placed in the muffle furnace at 50, 100, 150, 200, 250, and 300 °C. At 450, 550, 850 °C and other conditions, after heating and holding for half an hour, then reweighting, the loss on ignition of the paper at different temperature was calculated. The mass loss ratio of the same sample at different temperatures is shown in Table 2.

**Table 2.** Mass loss ratio of the side flower water transfer paper sample at different temperatures.

| No. | Roasted Flower Temperature (°C) | Loss on Ignition (%) |
|---|---|---|
| 1 | 50 | 1.79 |
| 2 | 100 | 2.06 |
| 3 | 150 | 12.4 |
| 4 | 200 | 53.82 |
| 5 | 250 | 60.6 |
| 6 | 300 | 60.8 |
| 7 | 450 | 61.3 |
| 8 | 550 | 61.5 |
| 9 | 850 | 61.5 |

Table 2 shows the loss on ignition and the flower paper weight percentage of the same sample at different temperatures. It can be seen from Table 2 that some volatile substances in the flower paper begin to volatilize when the temperature is continuously increased to 50 °C, and the volatilization in the early stage is slow, and the tobacco is roasted at the same time. Gas begins to produce. After the temperature rises to 150 °C, the volatile matter in the flower paper is largely lost, and the pungent odor in the roasted tobacco smoke becomes more and more obvious. Until the temperature reaches 250 °C, the loss of ignition in the flower paper reaches the highest value, and at high temperatures, almost no VOC and other organic and inorganic substances that can be burned out.

3.3.2. Loss on Ignition of Different Flower Papers at 850 °C

After studying the loss on ignition of the same flower paper at different temperatures, it was found that the quality of the flower paper began to be constant after 550 °C, and almost no VOC and other organic and inorganic substances that could be burned out. Through this conclusion, we continue to study the percentage loss of flower paper loss at different papers at 850 °C, and to study the estimated loss index of the amount of loss of paper on the total quality of the paper after the complete flowering. See Table 3 for the loss of paper at 850 °C.

The flower paper uses water transfer paper. The experimental results show that the loss on the flower paper after baking is about 60–80% of the quality of the flower paper. The possible reason is that the flower paper pattern occupies the size of the paper surface. The floral paper pattern is an inorganic oxide and is not easily decomposed at high temperatures. The pattern occupies a large amount of paper, and the loss on ignition is low, and vice versa. In the experiment, the loss of PVB paper accounted for about 80% of the flower paper, the water transfer paper was about 60%, and the low temperature paper was about 86%. The loss on ignition includes a portion of the VOC and other organic and inorganic materials that can be burned. Using a simple VOC meter, it is shown that as the temperature increases, the VOC concentration becomes larger, and the VOC concentration reaches the maximum when it is between 250 and 350 degrees. When the temperature rises again, the VOC decreases rapidly, indicating that it is in the flower paper. The organic matter was burned off.

**Table 3.** The loss of paper consumption as a percentage of the mass of paper indifferent papers at 850 °C.

| Name | Loss of Burning Content (%) |
|---|---|
| Social man (PVB) | 79.1962 |
| Social pig (PVB) | 80.3771 |
| Fu (word) | 75.8784 |
| Birds and flower | 74.3708 |
| Lion + child (side) | 73.8013 |
| Fujian Garden House | 86.2327 |
| Lion + child (up and down) | 67.8336 |
| Where the Yellow River meets the Yangtze River | 62.3425 |
| Fine pottery (small pieces) | 65.2784 |
| Peaceful and wealthy | 96.3303 |
| National color | 67.5095 |
| Side flower (water transfer paper sample) | 60.9921 |

*3.4. Effect of Temperature on the Loss of Wax in Special Ceramics*

Through the experiments, different types of special ceramics were covered under the cover of alumina powder. The waxing results of special ceramics were observed by slowly heating to 800 °C in a muffle furnace. Waxing results of special ceramic are shown in Table 4. Through the analysis of the results of special ceramic wax removal, among the special ceramics formed by hot die casting, paraffin wax, beeswax and oleic acid are used as binders, and the amount of VOC discharged is also different. The loss of wax in special ceramics is between 10–20%. According to the data, the amount of paraffin used in general hot die casting is 8–15%, beeswax is 0.5–1%, and oleic acid is 0.25–1%. The porcelain waxing process can basically complete the adhesive used.

**Table 4.** Waxing results of special ceramic.

| Name | Loss of Burning Content (%) |
|---|---|
| Talc porcelain 1 | 15.53 |
| Talc porcelain 2 | 14.24 |
| 95 porcelain 1 | 10.8 |
| 95 porcelain 2 | 11.25 |

*3.5. Accounting Analysis of VOC in Flue Gas of Roasted Flowers and Special Ceramics*

In this study, samples were taken from the flue gas and sent to Jiangxi MBM Environmental Testing Technology Co., Ltd. to determine VOC. The analysis results are shown in Table 5. The VOC amount and percentage of roasted flowers and special ceramic waxes calculated by the loss-of-burning experimental data are shown in Table 6.

The total amount of organic matter measured during the baking process of low-temperature flower paper was 26.6 mg, accounting for 3.17% of the flower paper, accounting for 3.96% of the loss on ignition. The amount of VOC in the water flowering paper (side flower) was 6.08 mg in the roasted flower experiment, accounting for 0.45% of the flower paper, accounting for 0.73% of the loss on ignition. PVB (Social pig) VOC was 8.36 in the roasted flower experiment, accounting for 0.92% of the amount of paper, accounting for 1.15% of the loss on ignition. Low temperature flower paper is the highest VOC (0.28%) content in the process of baking flowers used in the experiment. This may be related to the protection of the low-temperature flower paper for the protection of color-rich oxides, precious metals, etc., and because it belongs to low-temperature paper, the temperature of the roasted flower is 180–200 degrees, and the organic matter in the flower paper is greatly decomposed. Some of them become VOCs and are not decomposed into $CO_2$ and $H_2O$. Therefore, they have the most volatile organic compounds and the largest amount of VOCs. In the process of wax removal, the amount of TVOC in special ceramics is 35.15 mg, the proportion in special ceramics is 0.046%, and the proportion of loss on ignition is 0.42%.

The content of benzene in organic flue gas of special ceramic wax is up to 0.263 mg/m$^3$. In actual production, it is necessary to control the discharge of benzene and benzene-containing substances. The amount of paper used and the amount of special ceramics used in the experiment are small, and the amount of flower paper used in the on-site production process is large, which will inevitably lead to more serious VOC pollution, and this requires strict control of VOC emissions during ceramic baking and waxing.

**Table 5.** VOC analysis results in sample flue gas (mg·m$^{-3}$).

| No. | 1 | 2 | 3 | 4 | 5 | 6 |
|---|---|---|---|---|---|---|
| VOC Name | 95 Special Ceramics | Water Transfer Paper | Low Temperature Flower Paper | Tartar Oil | Cover Oil | PVB Paper (Social Pig) |
| Acetone | 0.02 | 0.02 | 0.01 | 0.01 | 0.01 L | 0.01 L |
| Isopropanol | 0.006 | / | 0.002 | 0.002 | / | 0.002 |
| Hexane | 0.047 | 0.005 | 0.053 | 0.004 | 0.004 L | 0.018 |
| Ethyl acetate | / | / | / | / | / | / |
| Benzene | 0.263 | / | / | 0.013 | 0.005 | 0.007 |
| 3-pentanone | 0.002 | 0.002 | 0.002 | 0.002 | 0.002 | 0.003 |
| N-heptane | 0.009 | / | / | / | 0.006 | / |
| Toluene | / | / | 0.018 | 0.006 | / | / |
| Ethyl lactate | / | / | / | / | 0.0277 | / |
| Butyl acetate | 0.005 | 0.005 | 0.007 | / | 0.005 | 0.006 |
| Ethylbenzene | / | / | 0.034 | / | / | / |
| Para-/m-xylene | / | / | 0.062 | 0.091 | / | / |
| O-xylene | / | / | 0.062 | / | / | / |
| Styrene | / | 0.004 | 0.005 | / | 0.004 | 0.011 |
| Anisole | / | 0.004 | 0.004 | 0.005 | / | / |
| Benzaldehyde | 0.007 | 0.013 | 0.027 | 0.009 | 0.013 | 0.026 |
| 1-decene | 0.003 | 0.003 | 0.006 | 0.003 | 0.006 | 0.003 |
| 1-dodecene | / | / | 0.008 L | / | / | 0.008 L |
| 2-heptanone | / | / | 0.062 | / | / | 0.003 |
| 2-nonanon | 0.003 | 0.003 | 0.015 | 0.004 | 0.008 | 0.003 |
| Propylene glycol monomethyl ether acetate | 0.005 | 0.005 | 0.007 | / | 0.005 | 0.006 |
| TVOC | 0.370 | 0.064 | 0.280 | 0.252 | 0.082 | 0.088 |

**Table 6.** VOC accounts for the weight proportion of flower paper, special ceramics and smoke gas.

| No. | 1 | 2 | 3 | 4 | 5 | 6 |
|---|---|---|---|---|---|---|
| Name | 95 Ceramics | Water Transfer Paper | Low Temperature Flower Paper | Tartar Oil | Cover Oil | PVB Paper |
| VOC in loss on ignition (%) | 0.42 | 0.73 | 3.96 | 3.2 | 0.97 | 1.15 |
| VOC in sample weight (%) | 0.046 | 0.45 | 3.17 | 3.2 | 0.97 | 0.92 |

## 4. Conclusions

The loss of burning paper in the process of baking flowers is between 60% and 80% of the total mass of the flower paper. The difference between the loss on ignition and the VOC content in the flue gas is relatively large, which is related to the amount of inorganic pigment used in the flower paper. If the amount of inorganic pigment per unit area is large, the proportion of loss on ignition is small, and the corresponding VOC content is small. At the same time, the VOC content of low temperature paper in the experiment accounted for the largest proportion of flower paper and loss on ignition, reaching 3.96%, followed by PVB paper at 1.15%, and VOC in water transfer paper accounted for 0.73% of loss on

ignition. The special ceramics formed by hot die casting will generate a large amount of flue gas in the wax discharge process, and the loss on ignition accounts for about 10–20% of the special ceramics. At the same time, the VOC content in the flue gas is generally around 0.4%. The amount of paper and special ceramics used in the experiment are small. The amount of paper and special ceramics used in the on-site production process are large, which will inevitably lead to more serious VOC pollution. It is necessary to control VOC emissions during ceramic baking and wax removal.

**Author Contributions:** Conceptualization, Y.C. and J.J.; methodology, Y.C., J.S and R.G.; validation, Y.C., J.J. and L.X.; formal analysis, nvestigation and data curation, Y.C.; J.J. and L.X.; writing—original draft preparation, Y.C. and J.J.; writing—review and editing, Y.C.; project administration and funding acquisition H.X. and C.Y. All authors have read and agreed to the published version of the manuscript.

**Funding:** This work were funded by the Environmental Engineering Evaluation Center, Ministry of Environmental Protection.

**Institutional Review Board Statement:** Not applicable.

**Informed Consent Statement:** Not applicable.

**Acknowledgments:** The authors are grateful to the National Engineering Research Center for Domestic and Building Ceramics, JCU and Jiangxi MBM Environmental Testing Technology Co., Ltd. for the assistance in analytical measurements.

**Conflicts of Interest:** The authors declare no conflict of interest.

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
