# Peer review of "Investigation and Accounting Research of VOC in Daily and Specialty Ceramic Industry"

_coatings, doi:10.3390/coatings12020279_

Round 1

Reviewer 1 Report

The manuscript entitled 'Investigation and accounting research of VOC in ceramic industry' lacks focus. The title is too broad and the authors failed to focus on a certain aspect.  The experimental section is not detailed either. We can predict the results since the experiment is too typical. The data (mainly FTIR and thermal analysis) are not good enough to draw a conclusion. We will need GC-MS data to understand these types of processes or mechanisms.

Author Response

Reply  to reviewer 1

Reviewer 2 Report

The manuscript describe a study on the VOC (volatile organic compound) emission from ceramic production. In the introduction, the authors should give a clearer context on the ceramic production as compared to other industrial sources in China such as what proportion of VOC emission from ceramic as compared to others. And where is the most concentration of ceramic production in China and its change over time. Is there any previous studies on the emission from ceramic production ?

The purpose of the study should be stated more clearly. And the implication of the study for VOC emission and air quality in China. There is little or no discussion in the Results and Discussion section.

Some specific comments are below

(1) Line 35-36: "..to win the battle against blue sky". Please clarify, use simple English

(2) Line 37: delete comprehensive in "the comprehensive management".

(3) Line 45: "functional" rather than "Functional"

(4) Line 46-47: Please give reference to "accounting for 60% of the world's total output; furnishings art porcelain accounts for 65% of the world, and specialty ceramics account for 45% of the world"

(5) Line 79-81: The authors should clarify the purpose of the study. Will the outcome of the study be used for emission inventory development or update the emission inventory. 

(6) Line 91: Change "plurality" to "number"

(7) Line 116: "Because the paper is thin and thick,..". The paper can not be both thin and thick. Do you mean "Because papers used can be thin or thick,.." ?

(8) Line 121: Replace "Calculation number" with the relation between the weight of papers before and after the baking is given by equation 1. Should have an equation number. 

(9) Line 193-194: Delete redundant sentence "Figure 3(a) shows the infrared spectrum of the cover, Figure 3(b) shows the infrared spectrum of the varnish." as the previous sentence already stated.

(10) Line 241: replace "can observe" with "shows"

(11) Line 256: "Figure 7 is" should be "Figure 7 shows"

(12)  Line 264: "Figure 7(b) whole process..." should be "In Figure 7(b), the whole process ..."

(13) In Table 5: what does the sign (/) mean ?

(10) Line 375: Delete "The" in "VOC The content"

Author Response

Reply to reviewer 2

Reviewer 3 Report

The manuscript presented is written in very poor English. The research scope is not clear and not clearly stated at the end of the Introduction, nor similar studies are discussed in the introduction.

The manuscript is more similar to a report than a research paper, e.g. in Table 2 and 3 redundant information is presented. Furthermore the figures have to be rearranged to be more informative and less in total number, keeping the more representative for the study and moving the others to the supplementary material.

Other example: at the end of Methods paragraph an "etc.." is written. Why? all the material and instruments have to be listed.

For the afore mentioned reasons I suggest a re-submission of the manuscript after a complete revision in all its parts.

Author Response

Reply to reviewer 3

Reviewer 4 Report

Manuscript coatings-1577979 is consistent with the objectives of the journal. There are no particular errors or omissions. An adequate background framework is provided and the objectives are well defined. The description of the methods used could be improved, but the results and overall discussion were clear. I suggest a spell-check of the text and I invite you to possibly consider the possibility of synthesizing some parts of the text. In addition to this, here are some comments that I ask you to consider:

  • Line 36: please check the meaning of this sentence “In order to win the battle against the blue sky”
  • “Equipment and instruments” Please provide further info on the used equipment and instruments (e.g., instruments characteristics) and please arrange this paragraph for better readability and understandability (i.e., "what was used to do what")
  • In table 1, pressure and volume are the same for all the considered test samples. Please consider to delete this two columns and report P and V in the table caption or in a footnote

Author Response

Reply to reviewer 4

Round 2

Reviewer 1 Report

The legends of the graph or figures could be made a bit bigger.

Author Response

I have   revised according to the reviewer1 again.

Reviewer 2 Report

The authors still do not address my questions

(1) In the introduction, the authors should give a clearer context on the ceramic production as compared to other industrial sources in China such as what proportion of VOC emission from ceramic as compared to others.

(2) And where is the most concentration of ceramic production in China and its change over time ?

Please address the above concerns

In addition, some minor comments on the revised manuscript

Line 82-84:

The sentence "The purpose of the study will the outcome of the study be used for emission inventory development or update the emission inventory." is unclear.

It should be "The purpose and outcome of the study will be used for emission inventory development or update the emission inventory."

Author Response

I have  revised according to the reviewer 2 again.

Reviewer 3 Report

The authors improved the introduction, but the few sentences added show that the manuscript needs a professional English editing. For example the sentence at lines 82-84 can barely be understood. Another paragraph that need an English revision is, e.g., par. 2.1, and so on.

Moreover I underline that the tables have to be updated keeping only the necessary information.

Table 2, 3, 4 and 6 seem to be simply a copy&paste from a lab notebook and they are not arranged in a proper way:

- Table 2: it appears that only columns entitled "Roasted flower temperature (°C)" and "Loss on ignition (%)" are informative, the others are redundant.

About Table 2 title: which sample?

- Table 3: keep only "Name" and and "Loss of burning content (%)" 

- Table 4: the same as for Table 3

- Table 6: keep only lines "Name", "VOC in loss on ignition(%)", "VOC in sample weight %)"

If the authors want to keep other listed data they have to explain why. 

Author Response

Ihave revised according to the reviewer3 again.
